# Scanning Probe Microscopy Techniques for Studying the Cell Glycocalyx

**DOI:** 10.3390/cells12242778

**Published:** 2023-12-06

**Authors:** Dmitry Kolesov, Anna Astakhova, Maria Galdobina, Alexey Moskovtsev, Aslan Kubatiev, Alisa Sokolovskaya, Leonid Ukrainskiy, Sergey Morozov

**Affiliations:** 1Moscow Polytechnic University, 107023 Moscow, Russia; 2Institute of General Pathology and Pathophysiology, 125315 Moscow, Russia; 3Mechanical Engineering Research Institute of the Russian Academy of Sciences, 119334 Moscow, Russia

**Keywords:** glycocalyx, scanning probe microscopy, atomic force nanoindentation, cell nanomechanics

## Abstract

The glycocalyx is a brush-like layer that covers the surfaces of the membranes of most cell types. It consists of a mixture of carbohydrates, mainly glycoproteins and proteoglycans. Due to its structure and sensitivity to environmental conditions, it represents a complicated object to investigate. Here, we review studies of the glycocalyx conducted using scanning probe microscopy approaches. This includes imaging techniques as well as the measurement of nanomechanical properties. The nanomechanics of the glycocalyx is particularly important since it is widely present on the surfaces of mechanosensitive cells such as endothelial cells. An overview of problems with the interpretation of indirect data via the use of analytical models is presented. Special insight is given into changes in glycocalyx properties during pathological processes. The biological background and alternative research methods are briefly covered.

## 1. Introduction

Living cells are in constant dynamic equilibrium with their environment. In this regard, the regulation of their interaction with the environment is an important task. Many components are involved in this regulation, with the glycocalyx being one of the most important. The glycocalyx covers most types of cells in the human body, including endothelial and epithelial cells, fibroblasts, and others. The glycocalyx serves a variety of functions depending on the cell type, including protecting the cell from mechanical damage, regulating cell adhesion and signaling, and participating in cell recognition and immune response. The structure and thickness of the glycocalyx can also vary between different cell phenotypes, reflecting their specific roles and functions. Since the discovery of the glycocalyx using transmission electron microscopy in the 1960s [1], interest in its structure and function has steadily grown.

The functions of the glycocalyx are determined by the complexity of its structure. Today, it is known to be a brush-like network of carbohydrates attached to membrane proteins on the cell surface. The transmission of biomechanical and biochemical stimuli into the cell is carried out with the help of transmembrane proteins that bind the glycocalyx and the near-membrane space inside the cell, called the cortex [2].

The glycocalyx is hydrophilic, which helps it to retain water molecules around the cell membrane and mediates many of its functions, and this trait certainly prevents the cell from drying out. The main components of the glycocalyx are glycoproteins and proteoglycans. Glycoproteins contain highly branched short carbohydrate side chains capped with sialic acids that are covalently attached to core proteins. Charged and highly branched carbohydrate chains regulate cells and the adhesion of proteins to the cell membrane. On the contrary, proteoglycans are decorated with long unbranched glycosaminoglycans (GAGs), such as heparan sulfate, chondroitin sulfate, and hyaluronic acid [3]. The GAGs of the glycocalyx are known to be predominantly negatively charged and, with the exception of hyaluronan, are covalently associated with the plasma membrane. 

The components are distributed in the glycocalyx in an extremely heterogeneous manner [4]. Two layers can be distinguished in terms of thickness and are enriched with various GAGs. There is also significant variation from the edges of the cell to the central part. Relatively recently, the ultrastructural organization of the endothelial glycocalyx and its relationship with the actin cytoskeleton were established. The dominant motif in it turned out to be a quasiperiodic structure comprising a hexagonal lattice with a distance of 20 nm between the core proteins, anchored in cellular structures that each have a diameter of about 10 nm; without taking into account the chains of polysaccharides and GAGs that form bushy structures, the latter form a typical step of 20 nm [5]. It is hypothesized that this lattice of ordered glycocalyx fibers can function as a molecular sieve. It should be noted that the glycocalyx is a very dynamic structure, and variations in the ionic strength, pH, and compositions of the cations lead to conformational rearrangements of the glycocalyx polymers and changes in their interactions.

The reported glycocalyx thickness varies from 200 nm to several microns [6]. Such a wide range is associated with a significant difference in the thickness of the glycocalyx for different cell types, locations in different organs and tissues of the body, and the measurement methods. Following the pioneering work of Luft [1], transmission electron microscopy (TEM) became a powerful instrument in the study of the glycocalyx. Different preservation and staining techniques, as well as cryomethods, have been developed [7,8,9,10]. The glycocalyx can be imaged via fluorescent or confocal microscopy using fluorescently labeled lectins or specific antibodies that are active against individual glycocalyx components [11]. High-resolution optical methods like STORM enable us to visualize the thin structure of the glycocalyx [12]. 

All of these studies make it clear that all sample preparations have significant influence on the measured characteristics, so the best way to study the glycocalyx is using unmodified cells under in vitro or ex vivo conditions. Scanning probe microscopy has long been successfully used under these conditions to study cells and individual macromolecules. Recently, ex vivo approaches using SPM have been introduced [13]. These not only can be used in imaging, but also provide unique information on the nanomechanical properties of the glycocalyx. Since Sun et al. [14] first pointed out the role of glycocalyx properties in observed “tethers” structures during atomic force spectroscopy experiments, a number of relevant papers concerning glycocalyx nanomechanics have appeared. Here, we present a review of the wide range of scanning probe microscopic methods that can be applied to the study of the glycocalyx under different conditions.

Given the variation in the glycocalyx’s functions and importance, it is not surprising that it is involved in a number of pathological processes. The glycocalyx has a direct relationship with endothelial dysfunction since it is involved in the regulation of NO production [15]. The degradation of the glycocalyx layer leads to increased cell permeability and the overtransport of water and macromolecules into the cell cytoplasm. It was shown that in rat myocardial capillaries, hyaluronidase treatment instantly results in myocardial tissue edema [16]. The glycocalyx is involved in the processes of tumor progression and metastasis due to its barrier function, role in growth factor storage, cell adhesion, signaling, and mechanotransduction [17,18,19]. We therefore aim to provide particular insight into investigations using the glycocalyx as a diagnostic or therapeutic target using SPM techniques.

## 2. Variations in the Structures and Functions of the Glycocalyx for Different Cell Phenotypes

The endothelial cells of blood vessels have a particularly thick glycocalyx layer compared to other cell types due to the endothelial glycocalyx playing a crucial role in vascular regulation (mechanotransduction, control of vascular tone, and the regulation of vascular permeability), preventing the adhesion of blood cells and platelets to the endothelium [20]. The thickness of the endothelial glycocalyx is different throughout the vascular bed: it extends from 0.2 to 0.5 µm in capillaries and venules, to 2–3 µm in small arteries, and to 4.5 µm in conductance arteries [21]. The glycocalyx thickness differs in veins and arteries: it varies between 20 and 400 nm in veins and is around 500 nm in arteries on average, reaching up to 4.5 μm in the carotid artery [22]. A dynamic equilibrium is established on the surfaces of endothelial cells, which determines the thickness of the glycocalyx: a balance is achieved between the removal of its components via the blood flow due to shear stress on the endothelial cells, the intensity of the production of structural elements by cellular machines, and the attachment of glycosaminoglycans to the membranes of the endothelial cells.

The endothelial glycocalyx consists of glycoproteins (selectins, integrins), proteoglycans (principally syndecans and glypicans), and glycosaminoglycan (GAG) [23]. Heparan sulphate proteoglycans account for 50–90% of the total amount of endothelial proteoglycans [21]. Most interactions between syndecans and extracellular matrix molecules, cell adhesion molecules, and growth factors may be mediated by the electrostatic interactions of their heparan sulphate chains. Both heparan sulphate (HS) and chondroitin sulphate (CS) carry negatively charged sulphate groups and are covalently linked to transmembrane proteoglycans. The negative charge of the glycocalyx determines its interaction with the plasma components, repelling negatively charged molecules, as well as erythrocytes and platelets. Hyaluronan HA binds to the endothelial surface receptor CD44, providing the filtering properties of the glycocalyx molecular sieve [24], or intercalates throughout the glycocalyx, thereby contributing to hydration and viscosity [25]. 

Compared to the cardiac and pulmonary capillaries, cerebral capillaries have a thicker endothelial glycocalyx layer, which is better preserved following lipopolysaccharide (LPS) administration. The surface charges of the brain endothelial cells are more negative than those of other vascular endothelial cells. Compared to those in the lung and heart, the denser structure of the glycocalyx of the brain capillaries is associated with endothelial protection and may be an important component of the blood–brain barrier [26]. The area physically covered by the glycocalyx within the lumen of the cerebral capillaries is significantly larger than in the cardiac and pulmonary capillaries (40.1 ± 4.5% compared to 15.1 ± 3.7% and 3.7 ± 0.3%, respectively) [9].

Transmission electron microscopy showed an amorphously structured glycocalyx along the endothelial surface of both vein and lymphatic samples. The venous glycocalyx-like structure was thicker than that in human lymphatic samples (mean values of 66 ± 18 compared to 49 ± 11). Immunohistochemistry identified glycocalyx structures of glypican-1, mucin-2, agrin, and brevican (a proteoglycan of the peri-synaptic extracellular matrix) in human venous and lymphatic samples. Podoplanin was only detected in the lymphatic samples. Mucin-2 prevents defects in the intercellular junctions, a feature associated with the known functions of the lymphatic system in extracellular fluid homeostasis [22].

Epithelial cells, which are found in the tissues that line the body surface and cavities, also have a glycocalyx layer, although it is generally thinner than that in the endothelial cells. The glycocalyx helps to protect the epithelium from mechanical stress, provides a barrier function, and facilitates cell–cell adhesion. In epithelial cells, the glycocalyx consists of glycoproteins, proteoglycans, glycolipids, and mucins [27,28,29]. Mucins are glycoproteins with bulky O-linked glycan attachments that influence integrin function and cell signaling. The general functions of mucins on internal epithelial surfaces are to remove microorganisms and supply protective barriers. The thickness of the glycocalyx in epithelial cells typically ranges from 20 to 200 nm [7].

Conjunctival epithelial and corneal cells contain complex glycocalyx-enriched transmembrane mucins, proteoglycans, and glycosphingolipids [30]. Transmembrane mucins (MUC16) extend 500 nm from the plasma membrane, above other molecules present on the glycocalyx, thereby providing a physical barrier and masking receptors via steric hindrance. In the epithelial glycocalyx, MUC16 is reinforced by galectin-3, a multimerizing lectin that causes the carbohydrate-dependent crosslinking of transmembrane mucins. The ocular surface exposes host molecules that act as decoy receptors, impeding pathogen adhesion and the initiation of signaling cascades.

Neurons and muscle cells may also have a thin glycocalyx layer that participates in cell signaling and communication. The glycocalyx in neurons also includes glycoproteins, proteoglycans, and glycolipids. These molecules may play a role in cell adhesion, synaptic transmission, and cell signaling. The glycocalyces on the surface of neurons may interact with extracellular matrix molecules and other cell adhesion molecules to support proper synaptic connectivity and plasticity [31]. Their head groups protrude toward the extracellular space, significantly contributing to the cell glycocalyx, and in certain cells, such as neurons, they are major determinants of the features of the cell surface [32].

Most blood cells also have a specific glycocalyx on their surfaces [33,34,35,36] that is much thinner than in other cell types (5–30 nm). In immune cells, the glycocalyx is involved in cell adhesion, migration, and interactions with other immune cells. In platelets, the glycocalyx is involved in platelet adhesion, aggregation, and clot formation.

We have discussed the glycocalyces of the main tissue types: epithelial (including endothelium), connective, muscle, and nervous. The composition and structure of the glycocalyx can vary depending on the cell phenotype and its specific functions and can be modulated in response to various stimuli, such as inflammation or disease conditions.

## 3. Principles of Scanning Probe Microscopy Techniques

Scanning probe microscopy was invented in 1981 and represents a powerful instrument of materials science [37]. The introduction of the atomic force microscope (AFM) [38] allowed researchers to spread SPM techniques to biological samples. For this technique, the measuring element is a microcantilever, which is an ultra-precise force sensor. The sharp tip on the free cantilever end interacts with the surface and causes cantilever deflection, which can be measured via the readout system. Different signals, along with the topography, can be obtained to characterize the local adhesive, elastic, electrical, and magnetic properties of the samples. AFM makes it possible to study the surfaces of samples in both air and liquid, which is especially important for investigating biological objects [39]. Semicontact or non-contact modes are usually used for investigating biological samples due to their smaller impacts on the surface. These modes have more complex requirements to operate in liquid due to the viscosity effect on the resonant oscillations of the cantilever. As technologies develop, new techniques that are better suited to such delicate objects are emerging, such as peak force microscopy and ringing mode or scanning ion conductance microscopy (SICM). The peak force and ringing modes are subresonant tapping modes that are more oriented to the study of the adhesive, viscous, and elastic characteristics of samples. SICM is suitable for the investigation of biological samples in physiological liquid conditions based on its operating principle [40]. More careful consideration of SPM principles is beyond the scope of this review.

## 4. Sanning Probe Microscopy Imaging of the Glycocalyx

The glycocalyx is such a complex object that it is almost impossible to obtain its topography using common methods. Back in 1994, ref. [41] suggested that the presence of glycocalyx complexified the obtainment of a sub-micrometer resolution of the membrane surface for fixed and alive cells in an aqueous medium, with the sugar chains moving under the AFM tip during scanning. The authors used different enzyme treatments to partly degrade the glycocalyx. The best results were obtained with neuraminidase (30 mU/mL), which allowed the capture of 10–60 nm protein particles on the membrane surface. In [42], the authors visualized chitosan oligosaccharides (COS), which can bind to negatively charged glycocalyces on the surfaces of HUVEC cells, by means of electrostatic interaction. Moreover, in the topography signal, COS modification did not result in any differences in comparison to the untreated samples. Only in the lateral deflection signal, which is associated with the friction between the tip and surface, were COS-induced alterations at the cellular surface detected, indicating a modification of the endothelial glycocalyx. The applied loading force was minimal, suggesting that the increased friction was not related to a change in morphology, but rather to locally restricted alterations in the charge distribution at the outer cellular surface.

In [43], the authors used AFM for the visualization of changes in the surface of PC12 cells before and after enzymatic glycocalyx digestion (Figure 1). Imaging was performed in peak force mode within 24 h after fixation, increasing the risk of artefacts. Therefore, AFM was only used for visual illustration of the alteration of the glycocalyx structure, which correlated with the effects, as measured via other quantified methods. The peak force mode is based on the recording approach and retraction curves in each scanning point and thus represents an extension of the force spectroscopy method in the mapping format [44]. Due to the exceptional sensitivity of the microcantilever to force exertion, it becomes possible to distinguish between different regions in the surface deformation and highlight individual layers with pronounced elasticity.

The most promising regime of AFM for glycocalyx visualization is the “ringing” mode [45]. In this mode, cantilever damping oscillations (ringing signal) occurring after it disconnects from the sample surface in sub-resonance modes, such as peak-force tapping or force-volume, are under inspection. The processing of this signal provides up to five new channels of information. Among them, there is a disconnection distance that corresponds to the cantilever displacement when the last molecules, which are slightly adhered to the cantilever tip, are disconnected, and provides information regarding their length. The authors introduce the mapping of the disconnection distance in comparison to the height image of human melanoma cells obtained in the ringing and peak force tapping modes (Figure 2). Since the glycocalyx has the nature of a floating molecule network on the cell surface, it is very exciting to associate the patch structures observed on the cell surface with glycoproteins and glycosaccharides of similar heights.

## 5. The Study of Glycocalyx Nanomechanics Using AFM Techniques

The Young’s modulus of the glycocalyx differs significantly from that of the cell cortex. Thus, atomic force microscopy is widely used for investigations of the nanomechanical properties of the glycocalyx. Using this method, the cantilever tip indents the surface of a sample with a sharp or colloidal sphere probe. The force–distance curve builds the raw data (Figure 3). The application of atomic force microscopy for quantifying the mechanical properties of the endothelial glycocalyx was very clearly demonstrated in video format in [46].

AFM nanomechanic measurements allow us to not only obtain the glycocalyx stiffness but also evaluate the layer thickness. For this purpose, the obtained force curves should be analyzed analytically. Several models to interpret the force–distance curves have been developed in the literature. These models use different approaches to determine the structure and nature of the glycocalyx. One of the most simplified models represents the glycocalyx as a mechanical spring [47,48]. In this case, the glycocalyx’s response to tip indentation could be described using Hook’s law: F (x) = kx, where the proportionality coefficient k shows the glycocalyx stiffness. The key point is that the linear fit is determined for the initial part of the force curve from the contact point to the point of inflection, which is assumed by controlling the fitting factor R^2^.

Another approach uses contact mechanics that resemble the Hertz model of cell elasticity [49]. The Hertz model assumes the linear elasticity of the material. The applied force required to indent the cell body to the depth *d* is given by the following expression:(1)$$F=\frac{4}{3}E_{\gamma }\sqrt{R_{eff}}d^{3/2}$$
where *R_eff_* is the affective radius of a tip and a cell, $E_{\gamma }=\frac{E_{glx}}{1-\nu^{2}}$, and *E_glx_* and ν are the Young’s modulus and Poisson’s ratio of the glycocalyx, respectively. One can perform separate fitting of the force curve using a contact mechanics model for the initial soft glycocalyx layer and the more rigid cell cortex. However, the selection of the point of inflection remains arbitrary.

A more complex double-layered model approximates a cell covered with the glycocalyx as two layers with different Young’s moduli and a transition zone with a gradient change in elasticity [50]. The reduced modulus of the system becomes
(2)$$E=E_{glx}+(E_{cell}-E_{glx})\frac{P\xi^{n}}{1+P\xi^{n}},$$
and
(3)$$\xi =\frac{\sqrt{Rd}}{T_{glx}}{\left(\frac{E_{glx}}{E_{cell}}\right)}^{m}\left(\frac{1-B_{s}\nu_{c}^{2}}{1-B_{l}\nu_{g}^{2}}\right)\;\;$$
where *E_glx_* and *E_cell_* and *ν_g_* and *ν_c_* are the Young’s moduli and Poisson’s ratios of the glycocalyx and cell, respectively; *R* is the effective radius; *T_glx_* is the thickness of the glycocalyx layer; and *P*, *n*, *m*, *B_s_*, and *B_l_* are the constants determined from finite element fits. By fitting the experimental force curves, we can calculate the glycocalyx and cell stiffnesses and the glycocalyx layer thickness.

One of the most complex glycocalyx models, based on its polymer brush nature, was suggested by Sokolov et al. [51]. It uses the Alexander–de Gennes theory of polymer brush compression [52]. The model takes into account the simultaneous squeezing of the brush and the deformation of the cell body. The brush has two fitting parameters: the grafting density *N* and the brush length *L*, which describe the applied force as a function of the tip position:(4)$$F(h)\approx 100k_{B}TRN^{3/2}L{\cdot e}^{-2\pi \frac{h}{L}}$$
where *k_B_* is the Boltzmann constant, *T* is the temperature, and *R* is the effective curvature radius of a tip and a cell. The model could be used with regard to the limits of applicability, namely 0.1 < *h/L* < 0.8. This model was verified by the authors in many publications [53,54]. It was shown in [55] that, in contrast to the Hertz model, in the “brush” model, the elastic properties of the cell do not depend on the depth of indentation. Thus, the cell mechanics could be described using the effective Young’s modulus in a self-consistent way when the glycocalyx is considered to be a brush-like polymer structure. Further development of the model occurs upon highlighting two parts of the pericellular layer: the inner and outer sublayers [56]. The first one could be associated with membrane corrugations and protrusions, while the second one corresponds to the glycocalyx itself. The double-brush model allows us to separate the contributions of both sublayers, as well as the cell body, in the force curves.

In [57], the authors compared the models described in the same series of the experiments. They used the HAEC cell line, grown in a vascular cell basal medium supplemented with an endothelial cell growth kit. To modify the glycocalyx, heparinase–heparin treatment was used. Force indentation curves were obtained using a colloidal probe for untreated, heparinase-treated cells, and cells additionally treated with heparin. Then, the data were analyzed using four different models of the glycocalyx layer. The authors concluded that all of the models used showed consistent and qualitatively compatible results. The glycocalyx layer became shorter and stiffer after heparinase treatment, with the values partly being restored after heparin addition. On the other hand, the quantitative estimates showed slightly different values that were sometimes difficult to compare due to the features of the models. So, it is necessary to keep in mind all the assumptions and approximations in order to apply any of the models.

**Figure 3 cells-12-02778-f003:**
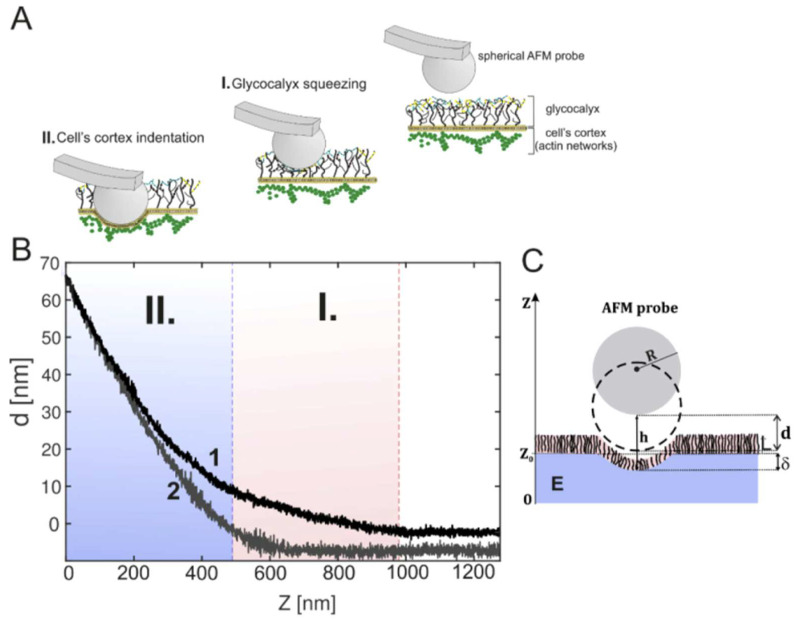
(**A**) Principal of atomic force nanoindentation technique for determination of glycocalyx nanomechanical properties; (**B**) example of obtained force curve with (1) and without (2) glycocalyx. Region I corresponds to glycocalyx layer while region II to cell body; (**C**) Definition of geometrical parameters. Taken from [57] with permission.

In [58], the authors also studied alterations in glycocalyx stiffness after treatment with a number of enzymes, heparinase I and III and hyaluronidase, via atomic force microscopy. The bovine lung microvascular endothelial cell (BLMEC) monolayer, cultured in vitro, was under investigation. A cantilever with a 18-micrometer glass microsphere was used not only for indentation but also for scanning the cell monolayer and assigning different regions in the layer morphology, that is, cell junctions or the cell nucleus’ location. During indentation, the authors observed a rapid increase in the Young’s modulus following the initial, almost plateaued segment. After enzyme digestion, the modulus increased more rapidly at smaller indentations. This effect was significantly stronger at the cell junctions and almost negligible at the nuclei region. To demonstrate the influence of the underlying cell structures on the measured stiffness, cytochalasin D was used to disrupt the actin filaments. In this case, the Young’s moduli increased much more slowly after the first 200-nanometer region.

This corresponds to the results obtained by Ke Bai and Wen Wang [59]. They investigated not only the spatial heterogeneity but also the temporal development of the endothelial glycocalyx on HUVEC cells. The AFM indentation data showed that the Young’s modulus of the cell membrane decreases over time from approximately 3 kPa on day 1 to approx. 0.3 kPa on day 14 of cultivation and beyond. Treatment with the neuraminidase enzyme significantly slows down and reduces the magnitude of the effect more than 6-fold. This point shows that the glycocalyx only fully develops after two weeks of cultivation. Moreover, the stiffness first decreases at the edge of the cell and then in the apical part. The results obtained were in good agreement with confocal imaging experiments.

Marta Targosz-Korecka and co-authors [60] investigated the morphological and nanomechanical changes in endothelial cells under stress conditions caused by either hydrodynamic flow or scanning with a colloidal probe. They showed that applied mechanical stress affects both the shape and polarization of the cells and their stiffness. Culturing HAEC cells in the flow condition for 48 h at a shear stress of 1.6 Pa causes the cells to have spindle-like shapes, with the main axis oriented in the flow direction (Figure 4A,B). The authors also observed the stiffening of the glycocalyx layer, while the thickness was almost the same (Figure 4C,D). This indicates that the glycocalyx apparently becomes denser under the flow conditions. Another picture was observed after scanning the cell surface with a colloidal probe in contact mode. The thickness of the layer decreased, while the Young’s modulus increased. The effect of scanning was similar to that of the treatment of endothelial cells with heparinase I; therefore, it can be concluded that the layer was damaged by the probe motion.

## 6. Examples of the Application of Nanomechanics to Studying the Glycocalyx in Pathophysiology

The glycocalyx plays a significant role in the signaling pathways of endothelium NO regulation. It is known that the degradation of the glycocalyx layer leads to a dramatic reduction in NO production by endothelial cells. However, the specific components of the endothelial glycocalyx involved in this response are not clearly established. The authors of [61] used an AFM cantilever tip modified with specific antibodies to selectively pull individual proteoglycans or glycosaminoglycans and detect changes in intracellular NO production. A sharp pyramidal tip was used to measure the adhesion forces of individual molecules, while tipless cantilevers were used to engage a potentially large area in the activation of NO production. In contrast, to measure the Young’s modulus, the retractive portion of the force curve was used. The mean adhesion forces for glypican-1, syndecan-1, CD44, heparan sulfate, and hyaluronic acid occurred in the range of 100–300 pN. The AFM pulling of glypican-1 and heparan sulfate for 10 min caused significantly increased NO production, whereas pulling on syndecan-1, CD44, and hyaluronic acid and with control probes did not (Figure 5).

The degradation of the glycocalyx and stiffening of the endothelium are important pathophysiological components of endothelial dysfunction. In [62], the authors studied these phenomena in tandem with diabetes by means of AFM nanoindentation. Nanoindentation experiments were performed using a colloidal spherical probe with a nominal diameter of 4.5 µm attached to the cantilever. Experiments were carried out ex vivo on prepared aorta samples resected from a descending thoracic fragment. The authors developed and verified a protocol to classify the obtained force curves into two types: regions that covered and not covered by the glycocalyx. This classification allowed significant heterogeneity in glycocalyx coverage at both the cellular and subcellular levels. The glycocalyx parameters (i.e., glycocalyx length and effective glycocalyx coverage) were obtained from indentation curves using the brush model proposed by Sokolov et al. [54]. The mean brush length of the glycocalyx was L = (674 ± 13) nm for the non-diabetic control mice. The authors also observed strong glycocalyx degradation in the diabetic mice. Interestingly, the endothelial stiffening and glycocalyx loss were present in early diabetes and remained similar at advanced stages, while the shrinking of the remaining glycocalyx correlated with diabetes progression (Figure 6). The authors note the importance of the preliminary classification of force indentation data before further fitting analysis.

Anne Wiesinger et al. [47] investigated the decrease in endothelial glycocalyx thickness and stiffness in experimental sepsis conditions using the atomic force microscopy (AFM) nanoindentation technique. A triangular cantilever with a 10 µm polystyrene sphere was used as a nanomechanical probe. Furthermore, 8- to 14-week-old male Lewis–Brown Norway rats and 8-week-old male mice were used for the ex vivo analysis of the aorta endothelial glycocalyx. Immunofluorescence microscopy of PECAM-1/CD31 markers was used to approve the preservation of the endothelial cell layer after sample preparation. The approach curves were analyzed. The authors distinguish two regions with different slopes in the force dependencies responsible for the glycocalyx (in the first several hundred nanometers) and cell cortex deformation. To prove that the AFM nanoindentation measurements were indeed able to disclose the endothelial glycocalyx, simultaneous AF and electron microscopy studies were performed on animals.

The AFM nanoindentation experiments showed that within 60 min after the infusion of heparinase I via the tail vein of rats, an approximately 50% (from 308 ± 68 nm to 154 ± 26 nm, *p* < 0.0001) decrease in glycocalyx thickness and a 33% (from 0.34 ± 0.05 pN/nm to 0.23 ± 0.01 pN/nm, *p* < 0.0001) decrease in glycocalyx stiffness were observed with respect to the control samples (n = 5 in each group). The same results (266 ± 17 vs. 137 ± 16 nm, *p* < 0.0001 and 0.34 ± 0.03 vs. 0.21 ± 0.01 pN/mn, *p* < 0.001) were observed after 18 h during the murine endotoxemia caused by the intraperitoneal injection of 1 mg/kg of BW lipopolysaccharide (LPS) from Escherichia coli into the mice. In vitro experiments conducted via the deterioration of the glycocalyx with several sepsis mediators, i.e., LPS, thrombin, and tumor necrosis factor-α alone, were also carried out. Further, the authors discuss the variations in glycocalyx thickness and stiffness identified by different researchers and via different methods. Studying the glycocalyx’s condition is significant due to its early manifestation in the pathogenesis of sepsis. However, it is not only glycocalyx degradation that occurs during the pathological process but also glycocalyx restoration, which is caused by antisepsis therapy, and this was investigated in [39]. The authors showed that cell treatment with Sulodexide (SDX), a heparin sulfate-like compound resistant to degradation by heparinase, leads to glycocalyx restoration. An AFM nanoindentation assay was used to demonstrate a significant increase in glycocalyx thickness. It was also found that SDX reduces vascular permeability, which is elevated in septic mice, and improves animal survival.

It is not surprising that cancer causes changes in the structure of the glycocalyx in tumor cells. Nadezda Makarova and coauthors studied the nanomechanics of epithelial bladder nonmalignant (HCV29) and cancerous (TCCSUP) cells with respect to the pericellular layer [63]. Precalibrated silicon nitride cantilevers with hemispherical probes were used to obtain the force–distance curves. The authors used the double-brush model to analyze the force curves, which provided information about the length and density of the pericellular layer. The inner brushes were found to be shorter for nonmalignant cells (210 ± 190 nm vs. 550 ± 290 nm) with the same density, while the outer layer was longer and denser (4.7 ± 4.2 μm vs. 1.8 ± 3.4 μm). Enzymatic treatments with heparinase I and neuraminidase were used to eliminate the two main components of the glycocalyx. The treatment had no significant effect on the inner layer, so it could be associated with the corrugation of the pericellular membrane. For the outer brush, heparinase I treatment significantly decreased both the length and density of nonmalignant cells and the length of cancer cells. In contrast, neuraminidase only has an increasing effect on the density of the outer brush of cancer cells. This corresponds with the fact that heparan sulfate is a major component of the glycocalyx. The removal of the sialic acid residues attached to the terminal ends of the surface glycans does not lead to significant alterations in molecule length, but it could affect the architecture via changes in the negative charge. In the absence of electrostatic stabilization, the long polysaccharide molecules stick together by forming a more rigid structure. The authors provide a schematic representation of differences in the structure of the pericellular layer for nonmalignant and cancer cells and their changes after enzymatic treatment (Figure 7).

We show data concerning the study of the glycocalyx nanomechanics under different pathological conditions in Table 1.

## 7. Other Scanning Probe Microscopy Methods for the Study of the Glycocalyx

Scanning ion conductance microscopy (SICM) is a relatively new technique of scanning probe microscopy in which an insulated surface can be examined using a nanoscale glass pipette in a conducting solution. SICM enables mapping of not only the topography but also the mechanical properties of living cells under physiological conditions with high spatial and temporal resolutions [66]. It was long thought that the SICM probe did not exert any stress on the cell surface during scanning because of the constant gap between the pipette tip and the sample controlling an ion current drop with the feedback system. However, the authors of [67] have shown that during experiments on the cell surface in “hopping mode” [68], the glass tip face repels the cell membrane before contact. Using built models, they found the presence of an initial soft region during the deformation of several glycocalyx-free cells, i.e., hippocampal neurons and the prion protein knockout cell line, apparently caused by the conformational slack in the cytoskeleton and its attachments to the plasma membrane. For cells with a glycocalyx, its thickness could be calculated by fitting the data obtained from the approach curves. For human mammary fibroblasts, it was estimated to be 70.5 ± 1.6 nm. It was shown that some subcellular features exhibit strong contrast in stiffness but none in topography, and the thickness of the glycocalyx should be taken into account for the chosen pipette tip aperture. 

## 8. Conclusions

Scanning probe microscopy is undoubtedly suitable for studying a structure as delicate as the glycocalyx. While the imaging of the glycocalyx is still problematic, measuring the mechanical properties produces unique information about its properties and structure under physiological conditions. In several publications, alterations in the nanomechanical properties of the glycocalyx were shown to be associated with a number of pathological processes characterized by endothelial dysfunction. Different approaches to data analysis allow us to appreciate parameters like the thickness, stiffness, and drafting density of the glycocalyx layer. However, researchers must be very careful when interpreting such indirect data. While atomic force spectroscopy has already become the classical method of studying the glycocalyx, new techniques like scanning ion conductance microscopy promise new perspectives for such investigations.

## Figures and Tables

**Figure 1 cells-12-02778-f001:**
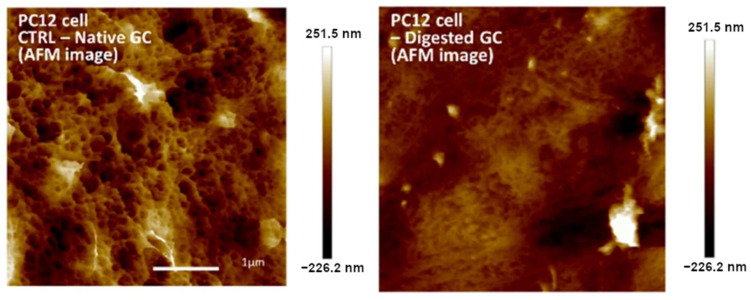
AFM images of PC12 cells obtained in peak force tapping mode. Left: surface of native PC12 cells; right: surface of PC12 cells after enzymatic removal of glycocalyx. Modified from [43].

**Figure 2 cells-12-02778-f002:**
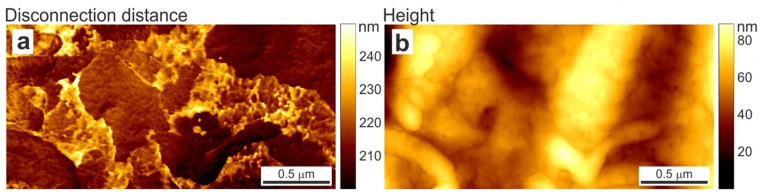
Distribution of (**a**) disconnection distance obtained in new AFM ringing mode and (**b**) height signal simultaneously recorded in peak force tapping mode on A375 human melanoma skin epithelial cells. Taken from [45] under Creative Commons license.

**Figure 4 cells-12-02778-f004:**
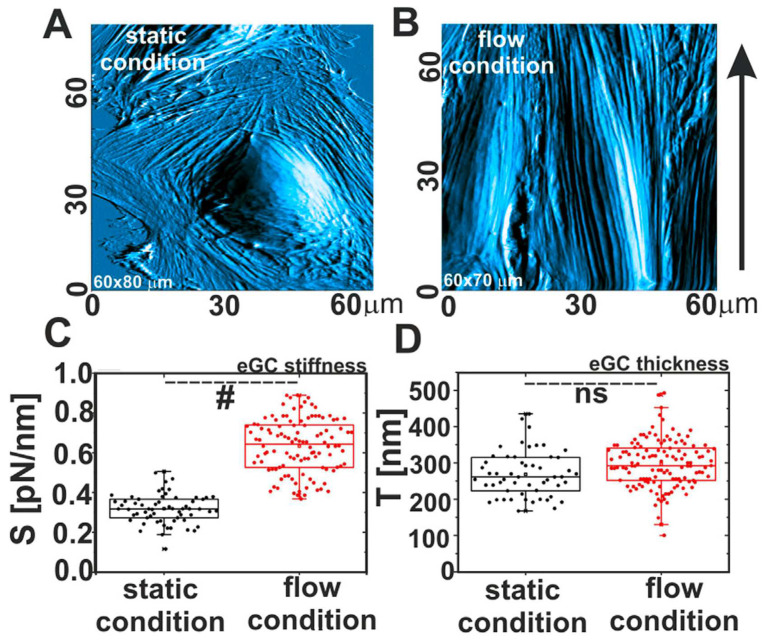
Comparison between morphological and nanomechanical properties of endothelial cells grown in static and flow conditions. (**A**,**B**) AFM images of cells, (**C**) glycocalyx stiffness, and (**D**) glycocalyx thickness. Statistics: (#) different at *p* = 0.05, (ns) non-significant. Modified from [60].

**Figure 5 cells-12-02778-f005:**
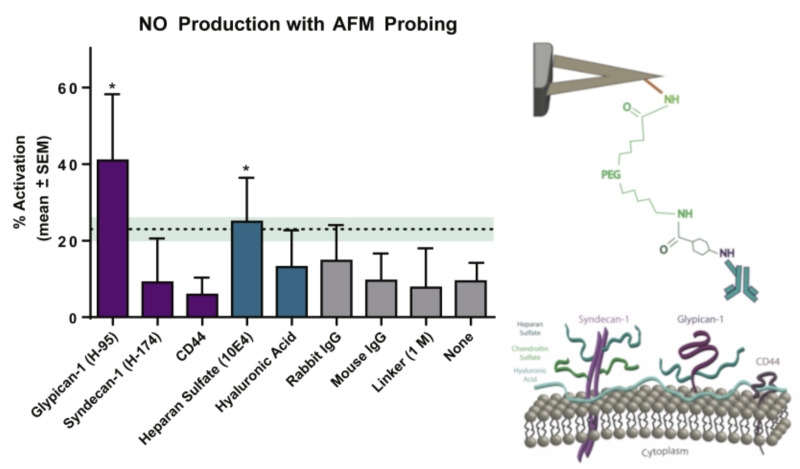
Alterations in NO production by rat fat pad endothelial cells after selective 10 min of pulling on the different components of the glycocalyx using the AFM probe. Right: scheme of cantilever functionalization. Statictics: (*) *p* < 0.05 compared to paired unstimulated regions Modified from [61].

**Figure 6 cells-12-02778-f006:**
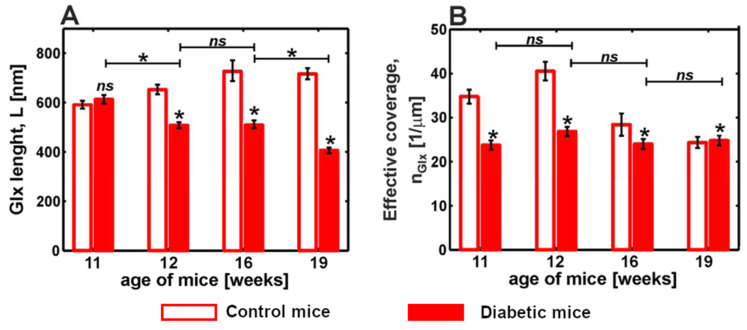
Comparison between glycocalyx length (**A**) and effective coverage (**B**) of endothelial cells in ex vivo mouse aorta with diabetes progression. Statistics: (*) *p* < 0.0001; (ns) non-significant. Modified from [62].

**Figure 7 cells-12-02778-f007:**
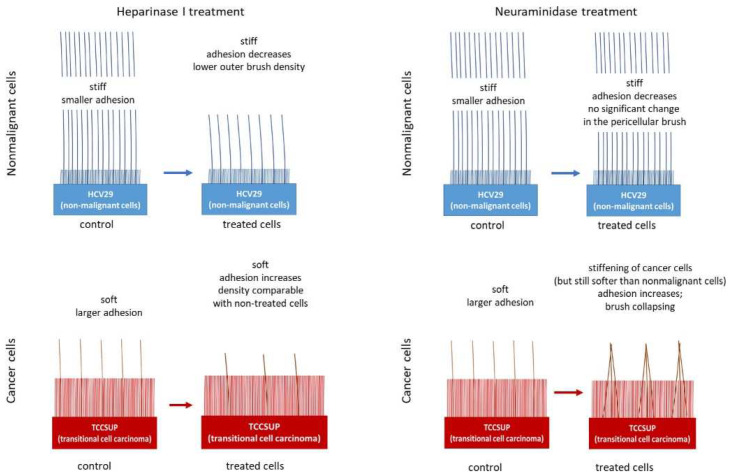
Schematic representation of the differences in the structure of the pericellular layer of nonmalignant and cancerous cells and their changes after enzymatic treatment. Taken from [64] under Creative Commons license.

**Table 1 cells-12-02778-t001:** The use of AFM methods for study of cell glycocalyx nanomechanics in pathological conditions.

Cell Type	Associated Disease	Method/Model Used	Short Description	Reference
Rat fat pad endothelial cells (RFPECs)	Endothelial dysfunction	Atomic force spectroscopy	The authors declare glypican-1 and heparan sulfate as the primary components of the glycocalyx that are responsible for stress-associated NO production	[61]
Endothelial cells on ex vivo aorta samples	Endothelial dysfunction and diabetes	Nanomechanics, brush model	Strong degradation of glycocalyx in diabetic mice	[62]
Endothelial cells on ex vivo aorta samples/in vitro HUVEC, EA.hy 926, HPMEC, bEnd.3, and GM7373 endothelial cells	Endothelial dysfunction in sepsis	Nanomechanics, mechanical spring model	A decrease in glycocalyx thickness after intraperitoneal injection of lipopolysaccharides. An in vitro decrease in glycocalyx thickness after heparinase I, thrombin, LPS, or TNF-α treatment	[47]
Bladder epithelial nonmalignant (HCV29) and cancerous (TCCSUP) cells	Cancer	Nanomechanics, brush model	The differences in the structures of the pericellular layers of nonmalignant and cancerous cells and their changes after enzymatic treatment	[63]
Brain microvascular endothelial cells (bEnd.3)	Endothelial dysfunction in sepsis	Nanomechanics, mechanical spring model	Restoration of glycocalyx thickness after Sulodexide treatment in cells pretreated with lipopolysaccharides	[64]
EA.hy926 endothelial cells and A549 lung carcinoma cells	Cancer and diabetes	Nanomechanics, brush model	Incubation in metformin leads to a significant increase in glycocalyx density and the length of endothelial cells and has almost no effect on lung carcinoma cells	[65]

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
