# Peer review of "Scanning Probe Microscopy Techniques for Studying the Cell Glycocalyx"

_cells, 2023, doi:10.3390/cells12242778_

Round 1
Reviewer 1 Report
Comments and Suggestions for Authors
Readability of the manuscript is very much hampered by poor grammar and syntax of the sentences.
In addition, not always were the proper references used
Comments on the Quality of English Languagesee, above
Author Response
Thank you very much for your response!
We ordered language editing by the MDPI language service. We hope, this will improve the readability of our manuscript. Also we have carefully checked all references.
Best regards,
Team of the authors.
Reviewer 2 Report
Comments and Suggestions for Authors
Nice overview of several AFM papers on glycocalyx, but would be nice to see more details of the results and technical challenges of the individual studies.
Comments on the Quality of English LanguageMany mistakes in English; please let the manuscript proofread and correct by a professional of English language.
Author Response
Thank you very much for your response!
We added some details to description of several papers and extended the scope of articles with some really high-end studies. Also we added table with summary of review results.
We also ordered language editing by the MDPI language service. We hope, this will improve the readability of our manuscript.
Best regards,
Team of the authors.
Reviewer 3 Report
Comments and Suggestions for Authors
The authors presented a review of this study of glycocalyx using atomic force microscopy.
As written, the review can be an introduction to this area for beginners. If to write for experts, it should be much longer and cover much broader topics. Nevertheless, the present manuscript has merit in itself. However, before this reviewer can recommend this manuscript for publication, there are a number of amendments that can be done to make this review stronger, as well as a number of inaccuracies that have to be fixed. For example, “Since glycocalyx regulates cell adhesion it has influence on cancer development due effect on tumor cell mobility and metastasis process.” (It looks like an overstatement; why?) Another example: “Since Sun et al. [15] first described observed structures as ‘tethers’, a lot of relevant papers concerning glycocalyx nanomechanics.” Here, the authors mixed up nanomechanics and force spectroscopy, see my later comments as well.
Glycocalyx on different cell phenotypes serves different purposes. This review will be stronger if the authors clearly separate glycocalyx on endothelial cells (well studied and the electron microscopy) from the same on other cell types, in particular, epithelial ones.
Figure 1, demonstrating an example of imaging go glycocalyx is definitely outdated, and more importantly, it just shows the change of the cell surface after a particular treatment. It is still a substantial assumption that the observed difference is only due to removal of particular glycocalyx molecules. These days, there are actually images of distribution of glycocalyx molecules of the cell surface, see for example, figure 3 in Scientific reports 7 (1), 11828.
Section 3 shows again some confusion between force spectroscopy and nanomechanics. Force spectroscopy is the analysis of the retraction adhesive force curve, which allows to restore the energy landscape of the molecules being put off by the action of the AFM probe. It is quite a large separate area of AFM research. This can easily be fixed by simply calling this analysis as nanomechanics.
Figure 2 shows a rather specific force curve showing the deformation of glycocalyx. It might be rather confusing because this force curve was not representative in the original paper – it was used only to define the concept of glycocalyx stiffness. Figure 3a of that paper [PLoS ONE 8(11): e80905] shows the example of the real force curves, which are quite far from the one chosen to be shown by the authors in figure 2.
The pre-factor in formula 4 should be 100, not 50. It was indeed used in the first papers by Sokolov’s group (following the original citations that had an error), but later was fixed.
The following statement is unclear and maybe confusing: “This model was further developed by the authors and showed good relevance in several publications on different cell types”. This model was shown to be the only self-consistent model from the contact mechanics point of view: Biophysical journal 107 (3), 564-575.
I feel that one very relevant reference was missing “Mechanical Way To Study Molecular Structure of Pericellular Layer”, ACS Appl. Mater. Interfaces 2023, 15, 30, 35962–35972. This is essentially a new step in the investigation of molecular structure glycocalyx through the study of nanomechanics.
In the conclusion, it makes sense to mention Ringing mode [Scientific reports 7 (1), 11828.] and [https://doi.org/10.1007/978], the mode that allows for direct imaging go glycocalyx, in the discussion of the other methods to study glycocalyx.
Comments on the Quality of English LanguageNothing serious.
Author Response
Thank you very much for your response and meaningful comments!
Individual members of the team of authors have extensive experience in the field of scanning probe microscopy in general as well as cellular and molecular biology. This was the motivation for writing of this review as an entry point for applying the accumulated experience to a new intriguing area for us and we are open to any helpful feedback in this regard. We tried to improve some phrases and provide references to confirm some statements.
1) Glycocalyx on different cell phenotypes serves different purposes. This review will be stronger if the authors clearly separate glycocalyx on endothelial cells (well studied and the electron microscopy) from the same on other cell types, in particular, epithelial ones.
We have add a large section devoted to a more detailed description of the structure of the glycocalyx and its differences for different cell types. Most of the reviewed papers relate to the study of the glycocalyx on endothelial cells. However, few cases of studying endothelial cells are also presented. This mainly concerns applications for cancer research. For better separation, we have added a table in the last section.
2) Figure 1, demonstrating an example of imaging go glycocalyx is definitely outdated, and more importantly, it just shows the change of the cell surface after a particular treatment. It is still a substantial assumption that the observed difference is only due to removal of particular glycocalyx molecules. These days, there are actually images of distribution of glycocalyx molecules of the cell surface, see for example, figure 3 in Scientific reports 7 (1), 11828.
We have corrected the figure description for better correspondence to the presented images. This images showing the cell surface before and after enzymatic treatment, which probably should digest components of glycocalyx. So we cannot fail to mention these images, since they also related to the study of the glycocalyx using AFM.
3) Section 3 shows again some confusion between force spectroscopy and nanomechanics. Force spectroscopy is the analysis of the retraction adhesive force curve, which allows to restore the energy landscape of the molecules being put off by the action of the AFM probe. It is quite a large separate area of AFM research. This can easily be fixed by simply calling this analysis as nanomechanics.
Thank you very much for this remark. Of course we understand that AFS primary related to the studying of adhesive forces and binding forces between individual molecules. We followed some authors of reviewed articles, which assigned nanomechanics measurements also to AFS. We agree that correct terminology must be used and have changed the calling this analysis.
4) Figure 2 shows a rather specific force curve showing the deformation of glycocalyx. It might be rather confusing because this force curve was not representative in the original paper – it was used only to define the concept of glycocalyx stiffness. Figure 3a of that paper [PLoS ONE 8(11): e80905] shows the example of the real force curves, which are quite far from the one chosen to be shown by the authors in figure 2.
We also used this curve to illustrate the principal of nanomechanics measurements. As I understand, the given curve related to a real measurement and represents the magnified part of Fig3a for individual curve. But authors have used rather simple and may be outdated model of “mechanical spring”, so this example indeed could be confusing. We changed figure to another, more relevant one.
5) The pre-factor in formula 4 should be 100, not 50. It was indeed used in the first papers by Sokolov’s group (following the original citations that had an error), but later was fixed.
Thank you, we fixed mistake in formula.
6) The following statement is unclear and maybe confusing: “This model was further developed by the authors and showed good relevance in several publications on different cell types”. This model was shown to be the only self-consistent model from the contact mechanics point of view: Biophysical journal 107 (3), 564-575.
We have changed this phrase and provided the validation of brush model based on the mentioned reference.
7) I feel that one very relevant reference was missing “Mechanical Way To Study Molecular Structure of Pericellular Layer”, ACS Appl. Mater. Interfaces 2023, 15, 30, 35962–35972. This is essentially a new step in the investigation of molecular structure glycocalyx through the study of nanomechanics.
Thank you for recommendation of this paper. It is very fresh and we have missed it because it was published after the pool of literature was selected. But we spend some time before understood that there probably is a mistake in this article. In the result section provided data on the length of the glycocalyx before and after enzymatic treatment in tens of microns. This values are slightly confused. Moreover, in the figure 6 state more relevant values. Probably decimal points are missed in the text.
8) In the conclusion, it makes sense to mention Ringing mode [Scientific reports 7 (1), 11828.] and [https://doi.org/10.1007/978], the mode that allows for direct imaging go glycocalyx, in the discussion of the other methods to study glycocalyx.
Thank you very much for this comment. This method really escaped from our scope. The mention of this method made our review much better.
Best regards,
Team of the authors.
Reviewer 4 Report
Comments and Suggestions for Authors
The work is written correctly and discusses achievements in glycocalyx research using the SPM method. However, this is a basic approach to the topic.
My major comments:
- to my knowledge, the authors lack experience in glycocalyx research, which makes it difficult to draw their own conclusions based on their own results.
- lack of a paragraph explaining the most important methodological aspects of research using AFM
-lack of a paragraph describing the differences in the structure of the glycocalyx of different cell types supported by appropriate citations
- there is a bit of a mix of works relating to cells and works relating to tissues.
For a work to be published, authors should improve its quality and the scope of cited works.
Author Response
Thank you very much for your response!
We tried to improve our work according to your comments and comments of other reviewers. Here are our answers:
- to my knowledge, the authors lack experience in glycocalyx research, which makes it difficult to draw their own conclusions based on their own results.
Yes, the authors have limited experience in studying of glycolyx using SPM methods, however, individual members of the team of authors have extensive experience in the field of scanning probe microscopy in general and cellular and molecular biology. This was the motivation for writing of this review as an entry point for applying the accumulated experience to a new intriguing area.
- lack of a paragraph explaining the most important methodological aspects of research using AFM
We have joint information about the AFM methodology and its application to the study of biological objects in a separate paragraph. In our opinion, a detailed consideration of the principles of AFM is beyond the scope of this review, because it is planned for publication in a special issue dedicated to scanning probe microscopy, and is also more oriented on biological applications then technical details.
-lack of a paragraph describing the differences in the structure of the glycocalyx of different cell types supported by appropriate citations
We have add a large section devoted to a more detailed description of the structure of the glycocalyx and its differences for different cell types
- there is a bit of a mix of works relating to cells and works relating to tissues.
Yes, the review presents some articles devoted to the study of the glycocalyx on ex vivo samples. However, all of them focus on measuring the properties of cells within a tissue, but not averaged tissue. Such works are of particular interest in view of possible cooperative cellular effects. We have added references to the origin of the samples, and also added a table with brief information to improve readability
Best regards,
Team of the authors.
Round 2
Reviewer 4 Report
Comments and Suggestions for Authors
I have no comments on the revised version of the manuscript